## Protocol for the Work Engagement and Well-being Study (SWELL): a randomised controlled feasibility trial evaluating the effects of mindfulness versus light physical exercise at work

Maris Vainre ,[1] Julieta Galante ,[2] Peter Watson ,[1] Tim Dalgleish,[1,3] Caitlin Hitchcock [1,4]

For numbered affiliations see end of article.

**Correspondence to**
Maris Vainre;
maris.vainre@mrc-cbu.cam.ac.uk

## ABSTRACT

**Introduction** Mental ill health is a major cause of disability. Workplaces are attractive for preventative interventions since most adults work; meanwhile, employers are interested in improving employees' well-being and productivity. Mindfulness-based programmes are increasingly popular in occupational settings. However, there is inconsistent evidence whether mindfulness interventions improve work performance and how effective mindfulness-based programmes are, compared with other interventions, in preventing mental ill health.

**Methods and analysis** In this online randomised controlled feasibility trial, an anticipated 240 employees will be randomised to either a 4-week light physical exercise course or a mindfulness course of the same duration (1:1 allocation). The primary outcome is work performance, measured using the Work Role Functioning Questionnaire. We aim to evaluate the acceptability, feasibility and procedural uncertainties of a randomised controlled trial in a workplace, calculate an effect size estimate to inform power calculations for a larger trial, and explore whether improved executive function and/or enhanced mental health could be potential mechanisms underlying the effect of mindfulness on work performance. Outcomes will be collected at baseline, postintervention and 12-week follow-up.

**Ethics and dissemination** Approval has been obtained from Cambridge Psychology Research Ethics Committee. (PRE.2020072). Results will be published in peer-reviewed journals. A lay summary will be disseminated to a wider audience including participating employers.

**Trial registration number** NCT04631302.

### Strengths and limitations of this study

► A randomised controlled trial to lay the foundations to investigate the mechanisms of mindfulness intervention underlying effects on work performance.
► The study employs a range of outcome measures, including self-reported measures and cognitive functioning tasks.
► This feasibility trial is not powered to detect significant effects, but rather to estimate effect size to inform design of a larger later-stage trial.
► Several feasibility outcomes will be collected to inform a later-stage trial.

Poor mental health is responsible for 44% of work-related episodes of ill health[4] and according to conservative estimates, is thought to cost the UK's economy £45 billion annually[5] or 2% of UK's Gross Domestic Product. To reduce this burden, a growing number of employers provide programmes to improve well-being and work performance.

Mindfulness-based programmes (MBPs) are increasingly popular at places of work. Mindfulness is typically defined as 'the awareness that emerges through paying attention on purpose, in the present moment and nonjudgementally to the unfolding of experience moment by moment'.[6] Practising such awareness is linked to reduction in symptoms of anxiety, depression and stress in community populations.[7 8] There is also evidence that mindfulness training could improve overall well-being,[8 9] life satisfaction[10] and quality of life.[7]

Mindfulness practice may yield workplace benefits beyond emotional well-being. It has been proposed that mindfulness improves work performance[11] and reduces the negative effects of multitasking.[12] Yet, there is little

## INTRODUCTION
### Background and rationale

Mental illness is a major cause of disability worldwide.[1] Much of the adult population is employed and spends about a third of their waking hours doing paid work.[2 3] The occupational environment is therefore an opportune setting for preventative mental health interventions.

evidence to support these claims. A recent meta-analysis concluded that work performance was rarely assessed in trials investigating the outcomes of MBPs.[13] When work performance was measured, wide-ranging operational definitions were used, for example, engagement,[14–18] motivation,[15] absenteeism[19] and presenteeism,[17–19] rate of errors[20] and burn-out.[21] Thus, estimating an overall effect is difficult.[13 22] Methods for measuring performance in higher education have less variability, yet there is no clear indication that offering mindfulness training to university students improves their academic performance.[23]

The mechanisms underlying any effect of mindfulness on work performance are also yet to be determined while two mechanistic pathways stand out that could explain such an effect of MBPs. First, positive effects of MBPs on mental well-being are well established,[7–9] and mental well-being is linked to better work performance.[24 25] Conversely, mental health problems decrease employees' performance,[26–28] particularly if these problems are poorly managed.[29] However, an indirect effect of MBPs on workplace performance via improved mental well-being has yet to be evaluated.

A second potential mechanism could be an improved cognitive control over mental activity, which allows one to prioritise current task-relevant goals.[30 31] There are three potential facets of cognitive control that may be improved by MBPs: (1) shifting, that is, the ability to switch between multiple tasks; (2) updating, or the ability to frequently refresh information in working memory to ensure a currently relevant record of information and (3) inhibition: deliberately hindering dominant or automatic responses that are irrelevant to the task at hand.[32] Improved cognitive control, in turn, may lead to better performance on workplace tasks.[11 33]

Mindfulness has been shown to have a small effect on cognitive control.[34 35] A recent meta-analysis analysing outcomes of randomised controlled trials (RCTs) measuring the effects of cognitive control in MBPs for healthy participants found a small overall effect of Hedge's $g=0.2$.[35] However, we know little about how these changes in cognitive control manifest in the workplace.[13] While mindfulness may improve performance on tasks closely related to the practice, such as counting breaths,[36] it may not extend to other tasks, such as those completed at work.[37]

Furthermore, to date, research has primarily focused on the impact of mindfulness on cognitive control over emotionally neutral information. Yet, much of the everyday mental activity that we seek to regulate is emotionally positive or negative.[38 39] In the two meta-analysis of MBPs' effects on cognitive control published to date,[34 35] only one identified study used emotional stimuli within an cognitive control task. This study reported a null-effect of meditation on cognitive control measured via an attention network test when comparing negative and neutral conditions.[40] It is important to note that this study[40] was likely underpowered.

At work, it is arguably beneficial to inhibit emotional thoughts (eg, worrying about a recent argument with your spouse) that are irrelevant to the task at hand (eg, writing a report). A reduced ability to inhibit internal emotional stimuli may interfere with our ability to maintain focus on workplace tasks. There is evidence that emotional stimuli inhibit cognitive control, when measured using the Stop-Signal Task.[41–43] As mindful meditation trains the ability to move away from thoughts and images of negative emotional valence, practising mindfulness may enhance cognitive control over emotional mental events.[44] It is therefore important to determine whether MBPs improve workplace performance via enhancement of cognitive control skills such as the ability to move away from negative stimuli (eg, worries about task performance) or to decentre from negative mental content[45 46] and refocus attention on the task at hand.[22]

Understanding the mechanisms underlying effects of MBPs on work performance would (1) help to design more targeted interventions, (2) improve our attempts to assess MBPs, by designing and selecting more stringent outcome measures and control interventions and (3) inform an understanding of for whom MBPs may be most effective, and in which context.[47]

## Objectives

Current literature suggests that MBPs could improve work performance through increased mental well-being and/or cognitive control over emotional material. In order to test this, we need to control for one of the two pathways. Both the MBP and light exercise have been shown to reduce stress, depression and anxiety,[48–50] however, only mindfulness is expected to improve cognitive control skills. We chose light exercises as a condition to help to distinguish between the different pathways through which work performance may improve.

A definitive RCT is needed to evaluate these potential mechanisms. However, methodological uncertainties and questions of acceptability and feasibility need clarification to inform the design of such a trial.[51–53] We aim to conduct a feasibility trial to clarify these uncertainties and complete a preliminary investigation of the relationships between mindfulness training, workplace performance and the proposed mechanisms of action.

This feasibility trial will:
1. Estimate the between-groups effect size for the effect of mindfulness, relative to a light exercise control condition, on our primary outcome of work performance, in order to inform power calculation for a larger trial.
2. Explore whether improved cognitive control and/or enhanced mental health could be potential mechanisms underlying the effect of mindfulness on work performance.
3. Assess the acceptability of the interventions and the study design by monitoring recruitment, retention and adherence to the course.
4. Determine procedural feasibility of a later stage trial by evaluating the willingness of the participants to be

randomised and other practical implications of running a RCT at a workplace.

## METHODS
This protocol follows the guidelines for RCTs set by the Standard Protocol Items: Recommendations for Interventional Trials (SPIRIT) 2013 statement[54] (SPIRIT checklist in online supplemental file 1). Initial consent taking started in November 2020. Participants, irrespective of the time they consent, received access to baseline measures from 23 February 2021. Data collection finished on 23 March 2022.

## Study design
This is a participant-level RCT. Employees will be randomly allocated in a 1:1 ratio to either of two parallel groups.

## Eligibility criteria
Eligibility to participate in this study will be self-reported. The employers who have agreed to participate in the study are local councils or education providers or trade either in the publishing, electronics or construction industry with employees in a variety of roles, mostly in desk-based occupations. Individuals can participate if they work for the employers taking part in this trial, are based in the UK, and are not currently on a long-term leave. We will recommend that a participant chooses not to join the study if they:
▶ Are currently suffering from severe periods of anxiety, depression or hypomania/mania.
▶ Are experiencing other severe mental illnesses.
▶ Have had a recent bereavement or major loss.
▶ Have already completed a mindfulness course or have meditated more than 10 hours in the past 10 years.

## Intervention condition: *Be Mindful* MBP
Participants in the Mindfulness condition will complete the *Be Mindful* prerecorded online course by Wellmind Media. It incorporates elements from Mindfulness-Based Stress Reduction[55] and Mindfulness-Based Cognitive Therapy.[56] Course materials and instructional videos are accessed through a website (http://www.bemindfulonline.com).

The 4-week course consists of 10 sessions led by 2 teachers, 1 female, 1 male. Participants are taught to use formal meditations (focusing attention on the practice of meditation) as well as informal mindfulness techniques, such as mindful walking and mindful eating. Daily homework includes a formal mediation practice with the assistance of video/audiorecordings (up to 30 min), and one or two informal exercises per day (see table 1 for an overview). Every week, participants receive emails motivating them to practice and informing them when the next module is available. As this is a feasibility examination for a pragmatic trial, no modifications to the procedures to maintain adherence to the intervention will be implemented.

| Condition | Intervention: be mindful | Control: light physical exercise |
|---|---|---|
| No of sessions in total | 10 | 28 |
| Online coursework frequency | Twice weekly | Daily |
| Typical session and its length | Self-paced. Includes videos (average of 3–4 min in total per session), self-reflection exercises and brief reading tasks. | Videos of 10–13 min. |
| Homework frequency | Daily | Daily |
| Typical assignment | A formal meditation (10–30 min) and shorter task such as journaling or noticing. The frequency of the latter varies from daily to once a week. | Using the exercises while taking brief breaks during the day. |
| Reminders to encourage practising | Four times a week | Four times a week |

**Table 1** Comparison of the intervention and control group

## Control condition: light physical exercise
The 4-week control condition involves light exercises aimed at increasing mobility, reducing stiffness, improving blood circulation, and avoiding pain or repetitive strain injuries that may result from sedentary or repetitive tasks common in office environments. The prerecorded exercises will include whole-body slightly aerobic exercises such as rotation of joints and stretching. This online course was developed by JG, a public health doctor, together with an expert in body posture.

The control condition course is designed to match the intervention condition in overall time commitment, and the frequency of interaction with the participant (see table 1). It replicates the encouraged use of short breaks throughout the workday to focus on well-being, as in the intervention condition.

## Data collection
Data collection will take place at baseline (T0), after the courses finish (T1) and 12 weeks after completing the courses (T3) (see figure 1). Additionally, a brief questionnaire will be sent to the participants each workday. Data collected at T1 will be considered as the primary endpoint of interest.

## Outcomes
### Feasibility, acceptability, and procedural outcomes
To determine feasibility of a later-stage trial, we will examine descriptive statistics to:
1. Estimate between-condition effect sizes:
   – For the primary outcome, to inform a power calculation for a later-stage trial.
   – For the cognitive control outcomes, to determine suitability of these measures to index mechanisms of interest.

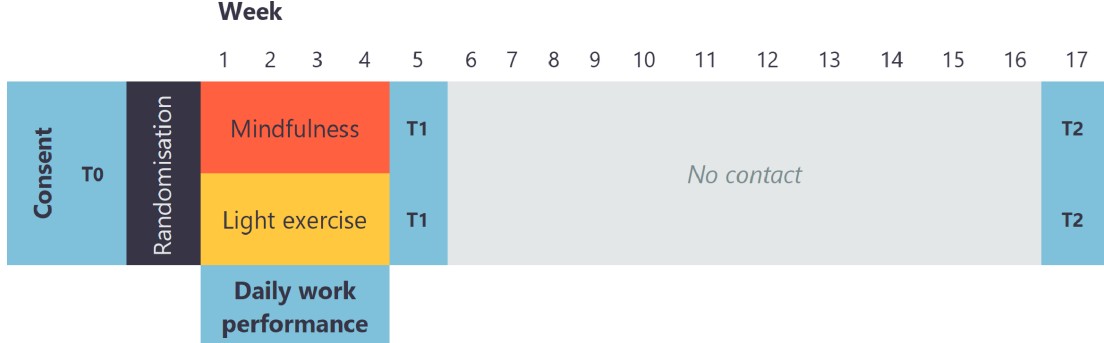

**Figure 1** Study procedures and timeline. Items in bold denote data collection.

2. Determine feasibility of running a later-stage trial by monitoring recruitment (the percentage of employees who consent into the study), retention, including timing of measurements (by indexing percentage of participants completing each time point), and potential contamination issues, most notably, measuring the extent to which participants talked about their course with participants from the other arm.

3. Acceptability of interventions by indexing which course the participants would have preferred to be randomised to, their regularity in engaging in exercise and mindfulness, and intervention dose, notably the percentage of course materials attempted.

4. Procedural uncertainties, for example, by exploring the suitability of our primary measure in indexing our primary outcome. To this end, we have introduced several work-related outcomes (see Secondary Outcomes).

5. Potential covariates influencing key outcomes which may need to be considered in design of the later-stage trial, including:
   - Participant mental and physical health at baseline.
   - Importance of job to participants' identity at baseline.

### Primary outcome: work performance

Work performance will be measured by using the 25-item Work Role Functioning Questionnaire's (WRFQ) [57] updated version,[58] to capture perceived difficulties in meeting work demands. Items are rated on a 5-point scale ('difficult all the time' to 'difficult none of the time'), with higher scores indicating better functioning. A sixth option denotes 'does not apply to my job'. The questionnaire has not been validated in English. Validations completed in Dutch,[58] Spanish,[59] and Norwegian[60] have shown good internal consistency (Cronbach alphas (α) 0.7–0.9),[58–60] and test-retest reliability (ICC=0.66, 95% CI 0.54 to 0.76 for the total score).[58] The WRFQ features four subscales: work scheduling and output demands (α=0.92), physical demands (α=0.92), mental and social demands (α=0.91), and flexibility demands (α=0.96).[58] The WRFQ has been shown to possess decent convergent validity, correlating with similar measures including the Utrecht Work Engagement Scale[61] (r=0.304),[58] Work Ability Index[62] (r=0.468).[58] The primary endpoint in this

trial will be the postintervention measurement. Feasibility of using the 12 week follow-up as the main outcome endpoint in the later-stage trial, will be assessed. We recognise that effects at 12 weeks may not be sustained longer term. Retention at 12 weeks will help to plan the sample size for a larger trial which could also then measure outcomes longer term.

### Secondary outcomes
#### Work-related outcomes

The participants are asked to report if they have health conditions that affect their ability to work, with options to pick one or several of the following: physical health problems, mental health problems, other health problems, no problems or prefer not to say. If a participant selects one of the first three options (ie, they have had problems), they will be asked to briefly describe these problems.

Those who self-report experiencing mental or physical health problems in the item described above will be asked to fill in the Work and Social Adjustment Scale.[63] The scale is widely used in the National Health Service psychology services in England and has good internal consistency, α=0.82.[64]

To get a better understanding of daily fluctuations that may occur in work engagement, participants will be asked to complete a 5-item version of the Work Role Functioning Questionnaire[65] each workday afternoon. Items are rated the same as in the full version of the questionnaire (see above).

#### Cognitive control mechanisms

Two online computerised cognitive tasks will be used to index our potential executive function mechanisms of interest. Affective cognitive control will be assessed using the Emotional Stop-Signal Task.[66] At the beginning of each trial within the task, a negative or a neutral image appears, followed by a go-signal (left or right arrow). Participants need to respond with a corresponding key-press. On a minority of trials (20%), the go-signal is followed by a stop-signal (upwards arrow) in which a go-response is required to be inhibited. Reaction times (in both, go- and stop-trials), response accuracy (failure or success in inhibiting response) and variability in reaction time throughout the task (a proxy for the ability to

overcome errors) will be measured. The main outcome of interest is the response time in stop-trials.

Participant's ability to track dynamic changes in their environment and alter their response strategies will be measured using an affective modification of the probabilistic reversal learning task.[67] The task will consist of six phases, three for the neutral and three for the negative condition. Each trial will begin with a negative or a neural image from the International Affective Picture System.[68] Next, pairs of stimuli (A-B or C-D) will be presented. Participants must select a stimulus with a key-press. In each pair, one of the stimuli is more likely to be rewarded (eg, selecting A or C is reinforced on 80% of trials). Feedback is presented after each response. Through trial-and-error, participants learn which stimuli are more frequently rewarded. After a certain number of trials (a phase), the contingency of reinforcement switches. In phase 2, the other stimulus in the pair is more frequently reinforced (eg, instead of A, B is now reinforced on 80% of trials). In phase 3, the reinforcement is switched again. Reaction times and response accuracy (ie, selecting the reinforced stimulus) will be recorded. The main outcome of interest will be changes in learning performance indexed via the proportion of correct responses.

### Other outcomes of interest
#### Well-being
Subjective mental well-being will be measured with the Short Warwick-Edinburgh Mental Well-being Scale (SWEMWBS), a seven-item questionnaire designed to capture a broad concept of well-being.[69] In SWEMWBS, items are scored on a scale of 1–5 ('none of the time'… 'all of the time'), with higher scores suggesting better mental well-being. The SWEMWBS internal consistency was $\alpha=0.84$ in a study in the UK general population (n=27 169).[70]

#### Stress
The Perceived Stress Scale (PSS) measures the extent to which the individual has perceived events as uncontrollable and overwhelming. The PSS consists of 10 items, answered on a scale of 0–4, higher scores indicate higher stress levels. The PSS possesses good internal consistency, $\alpha=0.84$–0.86.[71]

#### Depression
The Patient Health Questionnaire (PHQ-9)[72] is used to assess depression. It consists of 9 items answered using a scale from 0 to 3, and a further item asking about the level of difculty associated with any checked off items. Total scores range from 0 to 27 with cut-off points for depression at 5, 10, 15 and 20 for mild, moderate, moderately severe and severe depression, respectively.[72] A PHQ-9 score of at least 10 has been found to have 88% sensitivity and 88% specificity for major depression.[72]

#### Anxiety
The General Anxiety Disorder 7-item Scale (GAD-7)[73] assesses anxiety and has good reliability and validity.[74]

The items are answered using a scale from 0 to 3, yielding total scores between 0 and 21 with cut-offs at 5, 10 and 15 for mild, moderate and severe anxiety, respectively.[73] The scale's internal consistency is $\alpha=0.92$. A total score of 10 has a 89% sensitivity and 82% specificity for generalised anxiety disorder.[73]

### Mindfulness-related outcomes
The following will be administered to ensure that the MBP does increase mindfulness more than the control condition.

#### Decentring
The Experiences Questionnaire[45] is an 11-item measure of decentring. The items were generated to represent the changes believed to occur due to mindfulness practice, including the extent to which one's self-identity depends on one's thoughts, nonreactivity to negative experiences, and self-compassion. Statements are rated on a 5-point scale ('never' to 'all the time'), with higher scores reflecting higher levels of decentring. The scale's internal consistency is $\alpha=0.81$–0.84.[45]

#### Mindfulness
Mindful Attention Awareness Scale (MAAS)[75] is a self-report questionnaire consisting of 15 items designed to assess a core characteristic of mindfulness—a receptive state of mind in which attention simply observes what is taking place. Items are rated using a 6-point Likert scale ('almost always' to 'almost never'), with higher scores indicating more mindfulness. The internal consistency of MAAS is $\alpha=0.87$.[75]

### Sample size
One of the procedural uncertainties limiting the design of a fully powered trial is the size of the effect on the main outcome in interest. As a traditional power calculation is unfeasible given the lack of previous data, we seek to determine the likely effect size in this study, to inform a later-phase trial.

We aim to recruit 240 participants. A fully online design may cause a high loss to follow-up; a systematic review of internet-based RCTs found the average attrition rate to be 47% at postintervention.[76] Based on this, we have selected a sample size which we anticipate will yield complete data for 128 participants at our primary end-point of postintervention (64 per arm) and 68 participants at follow-up (34 per arm). In clinical research with lower attrition rates, feasibility trials tend to recruit 36 participants.[77] Considering high risk of attrition and the considerable uncertainties regarding the feasibility of the trial, we estimate that our sample size is optimal to examine the feasibility of procedures and provide a reliable estimate of effect size.

### Study procedures
#### Recruitment
Employers who have agreed to collaborate in the research project, have taken an active role in shaping the

recruitment process to their organisational customs. The invitation, sent via web-based communication services used by the employer, will have a link to the participant information sheet and consent form.

### Inducements for participation

There will be no inducements for completing either of the interventions. As a token of appreciation for completing the study assessments, participants will be given £10 at postintervention and £15 at follow-up time points in the form of retail vouchers.

### Randomisation procedure

After the participants have completed all baseline measurements, they will be randomised to either the mindfulness or the light physical exercise arm, stratified by employer. The randomisation process will be automated using Research Electronic Data Capture (REDCap), a platform for questionnaire data collection.[78 79] The allocation tables were generated with *randomizeR* package[80] in R using randomised permuted block randomisation with prespecified seeds for reproducibility. The code is available at GitHub.[81]

Participants will not be blind to their allocation. The primary analysis will be completed by a statistician (PW) blind to intervention allocation.

### Public involvement

The study's design has been formed by feedback from the employers participating in the study, including the perceived utility of the interventions, recruitment procedure and its timing, study materials, incentives for participation and outcomes. Changes to the initial design were proposed, some of which were implemented (eg, offering vouchers rather than cash; channels and timing for recruitment).

### Statistical methods

Central tendencies, dispersion and data missingness will be reported for all time points. At baseline, descriptive statistics will be presented overall and by group allocation. At following timepoints, outcomes will be reported by group.

Any significance testing, though not the focus of this trial, will follow the intention-to-treat principle. A key limitation of feasibility trials such as this is that adequate power is not obtained to detect statistical significance. For the primary outcome, a linear multiple regression model will compare the WRFQ total score between trial arms at postintervention, adjusted for baseline WRFQ and employer. Multiple imputation will be used to account for missing data. Further exploratory analysis will employ the same approach for other outcome measures at postintervention and 12-week follow-up. Mediation analysis techniques will be employed to assess the suggested mechanistic pathways. Effect sizes obtained in these analyses will be used to inform a potential later-stage trial and are the focus of this trial, rather than statistical significance.

For the secondary outcomes, mixed model repeated measures analysis will be performed using the daily monitoring of work performance to study changes between arms during the intervention. The analysis will also compare different work performance measures. Again, the focus of this trial is on obtaining an estimate of likely effect sizes, rather than statistical significance.

### Data monitoring and adverse events

An independent study steering group has been established to monitor data and advise the conduct of the study to ensure participant safety and integrity of research. We established the following safeguards[82 83]:
1. Participants were made aware they may request a consultation with a clinical psychologist if they feel uncomfortable with the study or experience discomfort they associate with the interventions.
2. Where participants' responses to PHQ-9 (depression) or GAD-7 (anxiety) are above clinical cut-off scores (≥20 and ≥15, respectively), a warning was automatically sent to the researcher (MV). For PHQ-9, the alert is also triggered when the participant score was >0 on the self-harm item. The researcher (MV) then consulted the clinical psychologist who contacted the participant.
3. Participants wishing to leave the study were encouraged to let the study team know why they have chosen to do so.

Any adverse events discovered through the mechanisms listed above were to be discussed with the Independent Study Steering group whose role was to decide whether they are attributable to the interventions (ie, adverse effects) and any subsequent course of action.

### ETHICS AND DISSEMINATION

The trial has received approval from the Cambridge Psychology Research Ethics Committee (PRE.2020.072).

### Consent

The consent form states eligibility criteria and the circumstances in which we recommend not to participate in the study. Participants are invited to join virtual information sessions or email the study team should they have any questions.

Information about accessing mental health support services is made available to anyone visiting the participation information website and e-mailed to those who consent to the study. Only those who consent to participate will receive the link to baseline measurements.

### Data management

Data will be collected and curated using the REDCap,[78 79] the Cohort Management System and JATOS.[84] Anonymised data will be shared for research purposes on request, in line with open science principles. All personally identifiable data will be separated from study data and stored on separate encrypted servers.

## Dissemination policy

Findings will be submitted to peer-review journals. Authorship in publications will be based on the International Committee of Medical Journal Editors' criteria. We will also send a lay summary of the results to the participating employers and participants.

### Author affiliations

[1]Medical Research Council Cognition and Brain Sciences Unit, University of Cambridge, Cambridge, UK
[2]Department of Psychiatry, University of Cambridge, Cambridge, UK
[3]Cambridgeshire and Peterborough NHS Foundation Trust, Cambridgeshire and Peterborough, UK
[4]School of Psychological Sciences, The University of Melbourne, Melbourne, Victoria, Australia

**Acknowledgements** The study team would like to thank Becky Gilbert, Camilla Nord, Marc Bennett, Rachel Knight, and Tierney Lee for their role in designing the cognitive tasks, Christina Haag for her consultations on planned analyses, and Markus Hausammann for substantially contributing to the design of the study logo.

**Contributors** MV, CH, JG and TD developed the idea for the trial and applied for funding. MV, CH and JG drafted the protocol which was then revised through discussions with TD, the Cognition, Emotion and Mental Health Programme lab at MRC CBU and the participating employers. The analysis plan was devised by MV, CH, JG and PW. MV is the lead researcher. CH and JG are supervising the research. The trial is sponsored by University of Cambridge.

**Funding** This study is supported by MRC Cognition and Brain Sciences Unit, MV is supported by Kristjan Jaak degree studies abroad scholarship by Estonia's Education and Youth Board, JG by the National Institute for Health Research (NIHR) Applied Research Collaboration East of England, CH by the Economic and Social Research Council and TD by the UK Medical Research Council (Grant reference: SUAG/043 G101400) and Wellcome (Grant reference: 104908/Z/14/Z, 107496/Z/15/Z).

**Patient and public involvement** Patients and/or the public were involved in the design, or conduct, or reporting, or dissemination plans of this research. Refer to the Methods section for further details.

**Patient consent for publication** Consent obtained directly from patient(s)

### ORCID iDs

Maris Vainre http://orcid.org/0000-0001-9570-3726
Julieta Galante http://orcid.org/0000-0002-4108-5341
Peter Watson http://orcid.org/0000-0002-9436-0693
Caitlin Hitchcock http://orcid.org/0000-0002-2435-0713

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
