## [Reviewer comments · BMJ Open]

ARTICLE DETAILS

TITLE (PROVISIONAL)	Protocol for the Work Engagement and Well-being Study (SWELL): A randomised controlled feasibility trial evaluating the effects of mindfulness versus light physical exercise at work
AUTHORS	Vainre, Maris; Galante, Julieta; Watson, Peter; Dalgleish, Tim; Hitchcock, Caitlin

VERSION 1 – REVIEW

REVIEWER	Jayanth Narayanan National University of Singapore
REVIEW RETURNED	06-Sep-2021

GENERAL COMMENTS	I have two comments/observations of the proposed study. 1. Use of physical exercise control: I see the need for an active control group for the study. However, I am not sure if Physical exercise if the appropriate control group. I recommend the authors to consider the treatments used by the following paper -
---

REVIEWER	Jantien van Berkel Wageningen University & Research
REVIEW RETURNED	01-Oct-2021

GENERAL COMMENTS	Congratulations on this well-written protocol of this interesting and relevant feasibility study and thank you for the opportunity to review. I have some questions, some minor textual remarks, and some suggestions for the final version of the protocol. First, the questions: in what (type of) organisations will this feasibility trial run? Employers have been involved in the protocol, but a description of the work setting seems to be lacking. This is however relevant, as the work setting for example for lorry drivers or construction workers is totally different than for office workers. This is also relevant for the feasibility of the larger trial. What do the authors consider to be the most important study limitations? There is a summary of strengths and limitations of the study, but a thorough discussion of these aspects I could not identify in the manuscript. Second, the minor textual remarks. Please note that a sentence in the abstract seems to be incomplete: " Mindfulness-based programmes are increasingly in occupational settings" (I can imagine something like 'used' or 'deployed' or ' applied' to be added there). Furthermore, one reference did not work: under data collection, line
---

	number 9. Please note that the SPIRIT statement was used, instead of the listed statements in the review protocol (CONSORT etc.). (Does not seem a problem to me, but I wanted to note it.) My suggestions relate to the design. I would suggest exploring whether a longer-term follow-up would be feasible, since effects are especially relevant in the long term. 16 weeks after baseline (4 weeks intervention + 12 week follow up after those 4 weeks) can hardly be seen as long-term effects. Secondly, I would recommend exploring whether it would be feasible to include data on process variables (such as reach, dose received, etc.) in your data collection. This would enrich insight in the feasibility. Since it is an online intervention, some of this might be done through collecting data on the use of the intervention; for example how minutes did participants spend on the intervention and their homework. Were all components used? Or maybe some skipped? Especially since this is the prelude of the larger study, gaining these insights can be considered relevant as they (also) play a role in feasibility.
--	---

VERSION 1 – AUTHOR RESPONSE

REVIEWER 1: DR. JAYANTH SINGAPORE

NARAYANAN, NATIONAL UNIVERSITY OF

#	Comment	Response
1	I have two comments/observations of the proposed study. 1. Use of physical exercise control: I see the need for an active control group for the study. However, I am not sure if Physical exercise is the appropriate control group. I recommend the authors to consider the treatments used by the following paper - *Comment from the Editor: We tried contacting the reviewer about how their comments should have finished, but unfortunately, we did not receive a response. Please respond to the reviewer's partial comment, if at all possible. Thank you	We agree that selection of a control group is nuanced, and have selected physical exercise on the basis that it is likely to control for non-specific treatment effects while not affecting our cognitive mechanisms of interest. We have now elaborated on this in the Introduction under Objectives. As data collection has now started, we are unable to change the control group, but will be sure to note the limitations of our selected control group in the trial paper.

REVIEWER 2

#	Comment	Response
1	Congratulations on this well-written protocol of this interesting and relevant feasibility study and thank you for the opportunity to review. I have some questions, some minor textual remarks, and some suggestions for the final version of the protocol.	We thank the reviewer for their supportive comments.

2	First, the questions: in what (type of) organisations will this feasibility trial run? Employers have been involved in the protocol, but a description of the work setting seems to be lacking. This is however relevant, as the work setting for example for lorry drivers or construction workers is totally different than for office workers. This is also relevant for the feasibility of the larger trial.	We agree that that this does impact feasibility for a larger trial. The types of organisations have been described under eligibility criteria. Since the first submission of the protocol to the BMJ Open, further employers have been recruited so we made changes accordingly.
3	What do the authors consider to be the most important study limitations? There is a summary of strengths and limitations of the study, but a thorough discussion of these aspects I could not identify in the manuscript	This study is impacted by all limitations relevant to a feasibility trial, and this has now been elaborated on (page 13); 'Any significance testing, though not the focus of this trial, will follow the intention-to-treat principle. A key limitation of feasibility trials such as this is that adequate power is not obtained to detect statistical significance.' The manuscript structure for trial protocols does not provide much space to elaborate on limitations, and we will be sure to do this thoroughly when presenting study results.
4	Second, the minor textual remarks. Please note that a sentence in the abstract seems to be incomplete: "Mindfulness-based programmes are increasingly in occupational settings" (I can imagine something like 'used' or 'deployed' or 'applied' to be added there).	Thank you for bringing this to our attention. We have now corrected this omission. The sentence now reads: "Mindfulness- based programmes are increasingly popular in occupational settings."
5	Furthermore, one reference did not work: under data collection, line number 9.	We think that this may be the link to Figure 1, which will need to be addressed in copy editing.
6	Please note that the SPIRIT statement was used, instead of the listed statements in the review protocol (CONSORT etc.). (Does not seem a problem to me, but I wanted to note it.)	We chose to use SPIRIT as that statement is designed for study protocols.
7	My suggestions relate to the design. I would suggest exploring whether a longer-term follow-up would be feasible, since effects are especially relevant in the long term. 16 weeks after baseline (4 weeks intervention + 12 week follow up after those 4 weeks) can hardly be seen as long-term effects.	We agree that effects after 12 weeks may not be sustained in the longer term and the duration of intervention outcome effects is important to establish. Although it is not possible to administer an additional follow-up with the current feasibility trial, this has now been emphasised as an important next step for a larger trial. This sentence was included in the primary outcome section: "We recognise that effects at 12 weeks may not be sustained longer term. Retention at 12 weeks will help to plan the sample size for a larger trial

		which could also then measure outcomes longer term.”
8	Secondly, I would recommend exploring whether it would be feasible to include data on process variables (such as reach, dose received, etc.) in your data collection. This would enrich insight in the feasibility. Since it is an online intervention, some of this might be done through collecting data on the use of the intervention; for example how minutes did participants spend on the intervention and their homework. Were all components used? Or maybe some skipped? Especially since this is the prelude of the larger study, gaining these insights can be considered relevant as they (also) play a role in feasibility.	These are very insightful and useful points, thank you. We have now added an exploration of dose to our feasibility aims, including examining the percentage of course materials attempted. Regarding reach, we will also report the percentage of company employees who consented into the study, which has now been added to the feasibility outcomes.

VERSION 2 – REVIEW

REVIEWER	Jantien van Berkel Wageningen University & Research
REVIEW RETURNED	22-Feb-2022
GENERAL COMMENTS	Congratulations on this interesting and relevant study protocol. My remarks and suggestions have adequately been addressed. Looking forward to the results. Please note the link to the figure (p8 line 49/50) is not correctly working yet.